# Immunotherapy in Melanoma: Recent Advancements and Future Directions

**DOI:** 10.3390/cancers15164176

**Published:** 2023-08-19

**Authors:** Meghan J. Mooradian, Ryan J. Sullivan

**Affiliations:** Massachusetts General Hospital Cancer Center, Boston, MA 02114, USA; rsullivan7@mgh.harvard.edu

**Keywords:** melanoma, immunotherapy, LAG-3, TIL, TLR-9, VEGF, vaccine, TIL

## Abstract

**Simple Summary:**

Progress in the treatment of malignant melanoma, particularly with the use of immunotherapy, has markedly improved outcomes. Despite these successes, many patients will not benefit from immunotherapy or will experience tumor growth after an initial response. Current efforts center on innovative therapeutic approaches to improve upon the advancements seen with single-agent programmed death receptor 1 (PD-1) inhibition and/or combination immune checkpoint inhibition. This review details several of the novel strategies under investigation in the treatment of patients with both resectable and unresectable disease.

**Abstract:**

Immune checkpoint inhibition has fundamentally altered the treatment paradigm of resectable and unresectable melanoma, resulting in dramatic improvements in patient outcomes. With these advances, the five-year overall survival in patients with newly diagnosed unresectable disease has eclipsed 50%. Ongoing research is focused on improving outcomes further, with a considerable emphasis on preventing de novo and acquired resistance and personalizing therapeutic options. Here, we review the ongoing advancements in the treatment of malignant melanoma, focusing on novel combination strategies that aim to build upon the successes of the last decade.

## 1. Introduction

Cancer immunotherapy was transformed by the identification of key immune checkpoints and the subsequent clinical translation of immune checkpoint inhibitors (ICIs). The first of these, ipilimumab, blocks cytotoxic T-lymphocyte-associated antigen 4 (CTLA-4) and was approved to treat advanced melanoma in 2011. In the 12 years since, the programmed death receptor 1 (PD-1) inhibitors pembrolizumab and nivolumab have each gained approval in patients with advanced melanoma, as has the combination of ipilimumab and nivolumab (ipi-nivo) (Figure 1). With these advances (Table 1), the five-year overall survival (OS) in patients with newly diagnosed unresectable disease has eclipsed 50%. Despite these successes, the majority of patients do not achieve a durable response to front-line ICI. The mechanisms of primary as well as acquired resistance to ICI are incompletely understood but include inadequate T cell infiltration into the tumor [1], immunosuppressive factors within the tumor microenvironment (TME) [1,2], loss of T cell function through the expression of alternative immune checkpoints [3], disruption of interferon gamma signaling via JAK1/2 mutations [4,5], impaired antigen presentation and neoantigen loss [5,6]. Efforts to overcome resistance are paramount and novel combination strategies are in development. This review focuses on the recent and ongoing advancements in the treatment of malignant melanoma that aim to build upon the successes of the last decade.

## 2. Treatment Advancements and Innovation

### 2.1. Lymphocyte Activated Gene 3 (LAG-3)

Lymphocyte activated gene 3 (LAG-3) is a cell-surface molecule that negatively regulates T cell activation [7]. PD-1 and LAG-3 are co-expressed on tumor-infiltrating CD4^+^ and CD8^+^ T cells, with pre-clinical data demonstrating the efficacy of dual blockade [8]. The RELATIVITY-047 study demonstrated the superiority of combination PD-1 inhibition plus LAG-3 inhibition over PD-1 inhibition alone for patients with treatment-naïve metastatic disease [9]. This phase II/III trial randomized over 700 patients with unresectable melanoma to single-agent nivolumab or the fixed-dose combination of nivolumab and relatlimab, an LAG-3-blocking antibody. Data demonstrate improved progression-free survival (median PFS: 10.1 months versus 4.6 months) as well as overall response rates (43.1% versus 32.6%) for the combination and, with 19.3 months of follow-up, a numerical improvement in OS, although this did not reach statistical significance (HR 0.80, confidence interval 0.064–1.01, *p* = 0.0593). Notably, grade ≥ 3 treatment-related AEs (trAEs) were more common with the combination, 21.1% vs. 11.1%, but less than that seen with the combination of ipi-nivo. Based on these published data, nivolumab and relatlimab (nivo-rela) received regulatory approval in March 2022, establishing this as a front-line standard of care option for patients with newly diagnosed, unresectable disease. In addition to nivo-rela, the anti-PD-1/LAG-3 combination of cemiplimab and fianlimab has exhibited efficacy, with phase 1 data (NCT03005782) demonstrating an ORR of 62.5% in treatment-naïve patients and 13.3% in those with prior anti-PD-1/PD-L1 exposure in the treatment of metastatic disease [10], with a sub-group analysis in a small cohort of patients (n = 13) demonstrating efficacy (ORR 61.5%) in those previously treated with PD-1 inhibition in the adjuvant setting [11]. Across cohorts, the safety profile of cemiplimab and finalimab was similar to that of cemiplimab monotherapy, with the exception of higher rates of adrenal insufficiency. This combination is being evaluated in a phase III randomized trial (NCT05352672) of patients with treatment-naïve unresectable melanoma, as well as high-risk resected (IIC-IV) cutaneous and mucosal melanoma (NCT05352672). 

### 2.2. Toll-Like Receptors (TLRs)

Toll-like receptors (TLRs) represent a family of pattern recognition receptors that recognize pathogen-associated molecular patterns (PAMPs) as well as damage-associated molecular patterns (DAMPs). TLR agonism results in the diverse modulation of the TME and immune cell activation [12]. Specifically, TLR-9 recognizes unmethylated CpG oligodinucleotides and, after activation, results in NFkB signaling and the release of type I IFN with subsequent CD8^+^ T cell activation [13]. Although no TLR-9 agonists are currently approved in the treatment of metastatic melanoma, several studies of TLR-9 agonists alone or in combination with ICI have demonstrated potential benefits, with an acceptable safety signal [14,15,16]. A phase Ib study of intralesional SD-101, a synthetic CpG oligonucleotide that stimulates TLR-9, in combination with pembrolizumab found the regimen to be safe, with an ORR of 78% in PD-1-naïve patients (7 of 9) and 15% in anti-PD-1-resistant disease (2 of 13), with responses seen in both injected and non-injected lesions. Treatment with SD-101 and pembrolizumab was associated with broad immune activation within the TME, including the infiltration of CD8^+^ T cells and induction of a type I INF response [15]. Tilsotolimod (IMO 2125), a TLR-9 agonist, has been studied in combination with ipilimumab. Though the phase I/II ILLUMINATE-204 study, which evaluated combination intralesional tilsotolimod and intravenous ipilimumab in anti-PD-1 refractory melanoma, demonstrated encouraging results, with an ORR of 22% and a disease control rate (DCR) of 77%, with a median duration of response (mDOR) of 11.4 months [14], the confirmatory phase III study of the ILLUMINATE-301 trial failed to demonstrate the benefit of combination therapy versus ipilimumab alone [17]. An additional TLR-9 agonist, vidutolimod (formerly CMP-001), has been studied in melanoma. Intralesional vidutolimod delivered alone (n = 40) or in combination with pembrolizumab (n = 159) was evaluated in a PD-1 refractory population, with data demonstrating a response rate of 17.5% in the monotherapy arm and 23.5% in the combination arm [16]. Correlative analysis illustrated that vidutolimod coupled with pembrolizumab was associated with an increase in tumor-infiltrating CD8^+^ T cells and PD-L1 expression in responding patients [18]. Combination vidutolimod and PD-1 inhibition has also been evaluated in the neoadjuvant setting, with encouraging efficacy data, specifically a major pathologic response rate of 60% in the 30 patients treated and an acceptable safety signal [19]. Further study of combination TLR-9 agonism +/− ICI is planned and/or ongoing in resectable and unresectable populations (NCT04935229). 

### 2.3. Vascular Endothelial Growth Factor (VEGF)

The inhibition of angiogenesis via vascular endothelial growth factor (VEGF) targeting has shown promise in the treatment of malignant melanoma. VEGF is a well-known mediator of immune suppression and modulator of tumor vasculature [20]. Correlative analyses from a phase I study of bevacizumab and ipilimumab in 46 patients with advanced melanoma revealed an activated endothelium with lymphocyte infiltration in tumors from patients treated with ipilimumab-bevacizumab in comparison with ipilimumab alone. Eight patients had the best response of a PR, 22 with SD, with grade 3–4 toxicity observed in 23.9% [21]. Notably, the benefit of VEGF inhibition coupled with ICI has been observed in mucosal melanoma, where the response to single-agent PD-1 inhibition is modest [22]. A phase Ib combination study of axitinib, an oral VEGF receptor tyrosine kinase inhibitor, with the PD-1 inhibitor toripalimab demonstrated an ORR of 48.3%, with OS rates at 1, 2, and 3 years of 62.1%, 44.8%, and 31.0%, respectively [23]. Based on these data, two randomized phase II trials comparing anti-tumor outcomes in patients with unresectable melanoma receiving ICI +/− bevacizumab were launched: NCT0195039, evaluating ipilimumab +/− bevacizumab in metastatic melanoma, and NCT04091217, evaluating the efficacy and safety of atezolizumab in combination with bevacizumab, specifically in patients with unresectable or metastatic mucosal melanoma. Interim analysis of this latter study demonstrated encouraging results in the first twenty-two patients, with an ORR of 36%, a DCR of 59%, and a median PFS of 5.32 months [24].

In addition to selective VEGF inhibition, the use of the multi-kinase inhibitor lenvatinib—a potent inhibitor of the VEGF receptors 1–2 as well as fibroblast growth factor receptors (FGFR) 1–4—in combination with pembrolizumab has been investigated. In the phase II LEAP-004 study, patients with unresectable melanoma with confirmed progression within 12 weeks of the last dose of PD-1/PD-L1 inhibition (including prior ipi-nivo) received lenvatinib 20 mg orally daily with pembrolizumab every 3 weeks [25]. The ORR in the PD-1 refractory population was 21.4%, with a median duration of response of 8.3 months and a 45.6% rate of grade 3–5 trAE, most commonly hypertension (21.4%)

### 2.4. Immune-Mobilizing Monoclonal T Cell Receptors against Cancer (ImmTACs) 

Immune-mobilizing monoclonal T cell receptors against cancer (ImmTACs) represent a new approach to immunotherapy. Tebentafusp (tebe), an ImmTAC, is a novel bispecific fusion protein that targets both the gp100 protein—which is expressed by melanoma cells—and CD3+ T cells, with the T cell receptor component restricted to gp100-HLA-A*02:01 complex recognition [26]. Engagement of the drug with the HLA-A*02:01 complex, which is present in 35% of African Americans and 50% of Caucasians [27], results in the activation of T lymphocytes. Tebe was recently approved in metastatic uveal melanoma based on the results of the IMCgp100-202 trial, which demonstrated an overall survival benefit of the fusion protein over investigators’ choice of therapy (single-agent pembrolizumab, single-agent ipilimumab, or dacarbazine), with OS at 1 year of 73% in the tebe arm and 59% in the control group [28]. Tebe has also been evaluated in metastatic cutaneous melanoma. A phase Ib trial demonstrated the safety and activity of tebe alone or in combination with escalating doses of the PD-1L inhibitor durvalumab and/or the CTLA-4 antagonist tremilumamb in an anti-PD-1 relapsed or refractory cohort. All combination arms demonstrated a 1-year OS of at least 75% [25]. Further study of tebe in PD-1 refractory cutaneous melanoma is underway in a randomized trial of tebe, the combination of tebe plus pembrolizumab, and an investigator’s choice arm that includes possible enrollment in other clinical trials (NCT05549297). 

### 2.5. Strategies to Mitigate Immune-Related Adverse Events (irAEs)

Recent emphasis has been placed on developing effective combination strategies that also aim to mitigate the known toxicity of ICI, specifically immune-related adverse events (irAEs), which can result in treatment delays, discontinuation, and the need for high-dose immunosuppression, which may dampen anti-tumor activity [29,30]. Based on pre-clinical data suggesting that interleukin-6 (IL-6)-Th17 pathway blockade may minimize irAE development without abrogating anti-tumor efficacy [31], combination ipi-nivo plus tocilizumab (toci) is under investigation. Preliminary data (NCT04940299) of combination ipi (3 mg/kg)-nivo (1 mg/kg) plus subcutaneous toci (162 mg/2 weeks for 6 doses) have been presented. Of 17 evaluable patients (12 with response data), 35% experienced a grade 3 irAE with a 58% ORR. Additionally, NCT03999749 is investigating “flipped dose” ipi-nivo with ipi (1 mg/kg) and nivo (3 mg/kg) in combination with toci (4 mg/kg IV every 6 weeks during the first 24 weeks). Early data demonstrate a favorable toxicity profile with a 25% rate of grade 3–4 irAEs and with evidence of activity (ORR of 70%). 

TNF-α inhibition is an alternative irAE mitigation strategy with intriguing pre-clinical and clinical data. TNF-α inhibition is a well-known management strategy for gastrointestinal irAEs [32,33,34], with pre-clinical data suggesting both irAE-protective and anti-tumor properties of anti-TNF [35,36,37,38]. TICIMEL, a two-arm phase Ib trial, evaluated 14 patients with unresectable melanoma. In the study, patients received ipilimumab (3 mg/kg) and nivolumab (1 mg/kg) combined with TNF-α inhibition with either infliximab (5 mg/kg, n = 6) or certolizumab (400/200 mg, n = 8). Both regimens were found to be safe (one instance of dose-limiting toxicity (DLT) in the infliximab cohort), with all evaluable patients (n = 7) in the certolizumab cohort experiencing a response, with a 50% ORR (3/6) in the infliximab cohort [39]. Further study of combination ICI and TNF-α inhibition is underway in a randomized phase II study (NCT05034536) of front-line PD-1 monotherapy or PD-1/LAG inhibition with and without infliximab (5 mg/kg for three doses) in the management of metastatic or recurrent melanoma. 

### 2.6. Oncolytic Viruses

Oncolytic viruses are novel agents that utilize naïve or genetically modified viruses to elicit an anti-tumor response. Talimonege laherparepvec (T-VEC) is a modified oncolytic herpes simplex virus type 1 (HSV-1) encoding granulocyte-macrophage colony stimulating factor (GM-CSF) and is the first and only approved oncolytic virus for the local treatment of unresectable melanoma. This approval is based on the phase III OPTIM study, which enrolled 436 patients with locally advanced or metastatic melanoma, randomized (2:1) to receive intralesional T-VEC or subcutaneous GM-CSF. Eligible patients had at least one cutaneous, subcutaneous, or nodal lesion and no more than three visceral lesions. The study met its primary endpoint, with a significantly higher durable response rate with T-VEC (19.3%) than GM-CSF (1.4%), with a favorable toxicity profile [40]. T-VEC was subsequently approved in the treatment of stage IIIB/C-IVM1a disease. With the single-agent activity of T-VEC, a study of combination therapy followed. The single-arm phase Ib Masterkey 265 trial assessed T-VEC in combination with pembrolizumab. In a small cohort of patients (n = 21) with unresectable stage IIIB/IVM1c melanoma, results demonstrated a CR rate of 43% and 4-year PFS and OS rates of 55.9% and 71.4%, respectively [41]. Based on these data, the phase III MASTERKEY-265 trial went on to evaluate the combination of T-VEC and pembrolizumab versus pembrolizumab alone. The combination was well tolerated; however, efficacy data did not demonstrate a significant improvement in PFS or OS for the combination [42]. 

Additionally, T-VEC has been evaluated in the neoadjuvant space, where the oncolytic virus (six doses pre-operatively) was associated with a 15% pCR rate and a 25% reduction in the risk of disease recurrence versus surgery alone in patients with resectable stage IIIB-IVM1a melanoma [43]. The 2-year OS was 88.9% for those receiving neoadjuvant T-VEC and 77.4% for those treated with resection alone, with the RFS and OS benefit maintained at 3 years. Combination neoadjuvant T-VEC and PD-1 inhibition with nivolumab is currently under investigation in the NIVEC trial (NCT04330430) [44]. In this single-arm phase II study, patients will receive four courses of intralesional T-VEC and three doses of nivolumab every 2 weeks, followed by surgical resection in week nine. The primary endpoint of this trial is the pathologic response rate.

### 2.7. Adoptive Cellular Therapies

Adoptive cell therapy (ACT) with tumor-infiltrating lymphocytes (TIL), first described in the 1980s [45], is a well-known form of immunotherapy that can recognize multiple tumor-specific neoantigens and has gained traction in the treatment of melanoma in light of several positive trials [46,47]. ACT is an approach where anti-tumor lymphocytes are harvested, grown, and stimulated ex vivo and then infused into a patient, often after receipt of a lymphodepleting cytotoxic regimen and along with vaccines and/or growth factors (e.g., high-dose interleukin-2) [48]. Early studies demonstrated encouraging results in small cohorts, with response rates of approximately 40–50%, many of them durable [45,49,50]. A recent phase II single-arm study examined the efficacy and safety of a single infusion of lifileucel (LN-144), an autologous TIL therapy, in a heavily pretreated population of patients with anti-PD-1 refractory unresectable melanoma [47]. Eligible patients received a nonmyeloablative lymphodepletion regimen, a single infusion of lifileucel, and up to six doses of high-dose interleukin-2 (IL-2). Of 66 evaluable patients, 36% (two CRs) obtained a response, many durable, with a median duration of response not reached. An additional randomized phase II study demonstrated the superiority of TIL over ipilimumab in patients, the majority of which had PD-1-resistant melanoma (86%), with an improvement in median PFS (7.2 months versus 3.1 months), ORR (49% versus 21%), and median OS (25.8 months versus 18.9 months). Notably, grade 3 or greater trAEs were experienced by all patients in the TIL arm, compared to 57% in the ipilimumab arm—with most events secondary to the conditioning regimen and IL-2—although quality of life scores were similar between treatment arms [46]. These data suggest that TIL therapy represents a significant improvement in a patient population with unmet need,, those resistant to PD-1 inhibitionIt is expected that lifileucel will gain FDA-approval based on the data described above.

### 2.8. Vaccination

Tumor-infiltrating lymphocytes have the ability to engage and irradicate melanoma cells in a human leukocyte antigen (HLA)-restricted manner through the identification of tumor-specific antigens. Melanoma antigens include overexpressed antigens on melanoma-specific tumor cells (e.g., glycoprotein 100 (gp100), melanoma antigen recognized by T cells 1 (MART-1)), cancer testis antigens (e.g., New York esophageal squamous cell carcinoma 1 (NY-ESO-1)), melanoma-associated antigen A3 (MAGE-A3) mutated oncogenes (e.g., *BRAF*, *NRAS*), and patient-specific mutated neoantigens [51]. Studies spanning the last two decades have demonstrated the ability to vaccinate with differing approaches, with varying degrees of success, against these classes of tumor-associated antigens (TAA) [52,53,54,55], with current efforts largely focused on the efficacy of combination vaccination and immune checkpoint inhibition.

In recent years, the study of individualized messenger RNA vaccines—personalized patient-specific vaccines based on the sequencing results of tumor samples—has shown encouraging results. BNT111 is an mRNA vaccine that encodes four melanoma-specific antigens (NY-ESO-1, MAGE-A3, tyrosinase, and transmembrane phosphatase with tensin homology). Data from the phase 1 dose escalation trial highlighted the ability of BNT111 to induce an immune response, with clinical responses accompanied by the induction of strong CD4^+^ and CD8^+^ T cell immunity against the vaccine antigens [56]. This agent is currently being explored alone and in combination with cemiplimab in patients with anti–PD-1 refractory and relapsed, unresectable melanoma (NCT04526899). In this trial, patients with tumors expressing at least one of these TAAs will be randomized to one of three intervention arms(2:1:1): BNT111 in combination cemiplimab, BNT111 monotherapy or cemiplimab monotherapy.

The phase IIb trial KEYNOTE-942 (NCT03897881) assessed the efficacy of mRNA-4157 (V940)—a single synthetic mRNA coding for up to 34 personalized neoantigens—in conjunction with pembrolizumab in patients with high-risk, resected stage IIIB-IV cutaneous melanoma [57]. Upon administration of the personalized cancer vaccine (PCV), the vaccine’s neoantigen sequences are taken up and translated by antigen-presenting cells, which can ultimately engage cytotoxic T lymphocytes and generate a tumor-specific response. Utilizing this technology, KEYNOTE-942 enrolled 157 patients with resected stage IIIB-IV cutaneous melanoma. Patients were randomized (2:1) to receive the PCV (I.M. every 3 weeks for up to 9 doses) plus 1 year of pembrolizumab (n = 107) or 1 year of pembrolizumab alone (n = 50). Relapse-free survival (RFS) was significantly improved in the combination arm, with an 18-month RFS rate of 78.6% in the vaccine arm and 62.2% in the monotherapy arm, with an acceptable safety signal. A phase III trial of the combination is scheduled to begin in the summer of 2023.

Immune-modulatory vaccines targeting known mechanisms of resistance to ICI offer a generalizable therapeutic strategy compared to patient-specific PCVs. A recent study evaluated a first-in-class immunomodulatory vaccine (IO102/IO103) against indoleamine 2,3-dioxygenase (IDO) and PD-L1, targeting immunosuppressive cells and tumor cells expressing IDO and/or PD-L1, combined with nivolumab. In thirty treatment-naïve patients with metastatic melanoma, the combination demonstrated compelling results, with an ORR of 80%, including a 43% rate of CR, a median PFS of 26 months, and a safety profile comparable to nivolumab monotherapy [58]. This agent has now been included in a registration-intent, randomized phase III trial (NCT05155254). An alternative immunomodulatory vaccine, mRNA-4329, which expresses IDO and PD-L1 antigens, is also under investigation with and without pembrolizumab in patients with advanced solid tumors, including melanoma (NCT05533697). Notably, these vaccination strategies differ from epacadostat, an IDO inhibitor, which did not improve PFS or OS when combined with pembrolizumab [59] in the management of front-line, advanced melanoma, as these are not strict IDO inhibitors but rather target IDO- and PD-L1-expressing cells.

### 2.9. Combination Immunotherapy and BRAF/MEK Inhibition

Activating mutations in *BRAF* (V600E/K) occur in approximately 50% of cutaneous melanomas, with these tumors vulnerable to BRAF/MEK inhibition [60,61]. The recent DREAMSeq data helped to shed light on the optimal front-line treatment for patients with newly diagnosed *BRAF*-mutant unresectable disease. This randomized phase III trial compared first-line ipi-nivo with first-line combination BRAF and MEK inhibition with dabrafenib and trametinib (dab-tram) in patients (n = 265) with treatment-naive, unresectable, stage III or IV melanoma with a BRAF V600E/K mutation. At a median follow-up of 27.7 months, 2-year OS rates (the primary endpoint) significantly favored first-line ipi-nivo (71.8% versus 51.5%), with data illustrating that ICI appeared less effective when used after targeted therapy [62].

An alternative strategy explored in *BRAF*-mutant disease is the use of *BRAF*-targeted therapy in concert with immunotherapy. The rationale for this combination regimen is based on evidence that oncogenic BRAF contributes to immune escape and that BRAF inhibition results in enhanced antigen presentation, a more favorable TME, and increased effector T cell tumor infiltration and activation [63,64]. IMspire 150 was a randomized phase III study evaluating atezolizumab plus vemurafenib/cobimetinib (vem-cobi) versus vem-cobi alone in patients with previously untreated *BRAF-V600*-mutant metastatic melanoma [65]. At a median follow-up of 18.9 months, PFS (15.1 months versus 10.6 months) and the median duration of response (21.0 months versus 12.6 months) were prolonged with atezolizumab versus placebo. At 2 years, the OS rate was 60.4% and 53.1%, respectively, with rates of trAE and treatment discontinuation similar between arms. Based on these positive data, the triplet combination of atezolizumab plus vem-cobi was approved in 2020. Alternative triplet approaches with combination anti-PD-1 and BRAF/MEK inhibition have also been studied, with mixed results. KEYNOTE-22 was a randomized phase I/II trial comparing pembrolizumab plus dab-tram with dab-tram alone in patients with previously untreated *BRAF-V600* mutation-positive advanced melanoma. Although the primary endpoint of PFS did not show a statistically significant improvement, after 9.6 months of follow-up, numerically higher values were observed in the triplet arm, 16.0 months vs. 10.3 months (HR, 0.66; *p* = 0.043) [66]. After additional follow-up, at a median of 36.6 months, mature results did show a clinically substantial improvement in the duration of response and survival [67]. Notably, rates of toxicity were much higher in the combination arm, with a 58% rate of grade 3–5 trAEs (one grade 5 event) versus 25% in those receiving targeted therapy alone. COMBI-I, which assessed the efficacy of the PD-1 inhibitor spartalizumab plus dab-tram in a similar population, was negative, with the triplet failing to significantly improve PFS [68]. In this phase III study, patients with *BRAF*-mutant unresectable melanoma were randomized to the PD-1 inhibitor spartalizumab plus dab-tram or placebo plus dab-tram. Although there was a numeric improvement in median PFS (16.2 months vs. 12 months), this did not reach statistical significance, with higher rates of grade ≥ 3 trAEs in the spartalizumab-containing arm (55% versus 33%).

In summary, while there is a rationale for combined BRAF/MEK inhibition plus anti-PD-1/PD-L1, with three randomized trials demonstrating consistent improvements in progression-free survival (as well as increased toxicity) with the triplet therapy versus targeted therapy alone, only one of these trials achieved statistical significance and received FDA approval. Based on these varied results, coupled with strong efficacy data of ipi-nivo in the *BRAF*-mutant population and no comparative data with triplet therapy versus immunotherapy, the widespread use of this strategy has been limited.

### 2.10. T Cell Immunoreceptor with Immunoglobulin and Immunoreceptor Tyrosine-Based Inhibition Motif Domain (TIGIT)

The targeting of inhibitory immune checkpoints expressed on the surfaces of T cells and natural killer cells is an area of active interest. TIGIT, also known as T cell immunoreceptor with immunoglobulin and immunoreceptor tyrosine-based inhibition motif domain, is a well-known marker thought to activate inhibitory receptors in T cells and NK cells as well as T regulatory cells.

Pre-clinical data demonstrate that TIGIT is upregulated on tumor-antigen-specific CD8⁺ T cells and CD8⁺ TILs from patients with melanoma, and notably these TIGIT-expressing CD8⁺ T cells often co-express PD-1, leading to the hypothesis that dual TIGIT and PD-1 inhibition may be an effective therapeutic strategy [69]. Indeed, an encouraging efficacy signal was seen when the anti-TIGIT antibody vibostolimab was combined with pembrolizumab in the treatment of early-stage, resectable melanoma. In this phase I/II study, a single pre-operative infusion of vibostolimab plus pembrolizumab was more effective than the oncolytic virus gebasaxturev plus pembrolizumab or pembrolizumab alone [70]. Although the pathologic complete response rate was similar between vibostolimab-pembrolizuamb (38%) and pembrolizumab alone (40%), the dual TIGIT/PD-1 inhibition strategy led to an improvement in ORR (50% vs. 27%), the 18-month EFS rate (85% vs. 78%), and the 18-month RFS rate (95% vs. 73%). Research on the combination is underway in the adjuvant setting in the phase 3 KEYVIBE-010 trial (NCT05665595), as well as in the advanced, PD-1 refractory setting, where the assessment of the combination of anti-TIGIT (vibostolimab) or anti-VEGF (lenvatinib) with anti-PD-1 (pembrolizumab) and anti-CTLA-4 (quavonlimab) is underway (NCT04305041).

### 2.11. Early-Stage Disease

#### 2.11.1. Adjuvant Strategies

With the success of ICI in unresectable, advanced melanoma (Table 1), focus has expanded to include the study of these agents in earlier-stage disease, with adjuvant single-agent PD-1 inhibition now approved in high-risk, resected stage IIB-IV disease [71,72,73,74] (Table 2). The approvals for single-agent pembrolizumab and nivolumab in stage III/IV disease were based on data from KEYNOTE 054 and Checkmate 238, which demonstrated an improvement in relapse-free survival (RFS) and distant-metastasis-free survival (DMFS) with the administration of PD-1 inhibition over a placebo [72] or CTLA-4 inhibition [73], respectively. Based on these positive results, adjuvant PD-1 inhibition was then evaluated in resected high-risk stage IIB/C disease. KEYNOTE 716 was the first trial to establish the benefit of adjuvant ICI in this population, with 1 year of pembrolizumab significantly improving RFS versus placebo (HR 0.61) [71]. The benefit of PD-1 inhibition over placebo in resected stage IIB/C disease was recapitulated in Checkmate 76K, with a 58% reduction in relapse risk or death for those receiving nivolumab versus placebo [74]. Based on these studies, adjuvant anti-PD-1 inhibition should be considered in patients with resected stage IIB-IV melanoma.

Aiming to build on the improvements in RFS seen with adjuvant PD-1 alone, investigators also examined the combination of adjuvant ipi-nivo. Although low-dose ipilimumab (1 mg/kg every 6 weeks) plus nivolumab failed to improve RFS versus nivolumab in patients with resected IIIB-IV disease in Checkmate 915 [75], ipilimumab (3 mg/kg) plus nivolumab demonstrated a striking RFS improvement in resected stage IV disease, with a 75% risk reduction versus placebo [76]. However, as with other studies, this improvement came at the cost of toxicity, with grade 3 and higher adverse events exceeding 70% in the combination arm. Ipi-nivo is not currently approved in an adjuvant setting.

Despite the clear improvement in RFS across adjuvant trials, with the exception of high-dose ipi (10 mg/mg) [77], at this time, adjuvant ICI with anti-PD-1 has not demonstrated a clear OS benefit. Whether this is due to insufficient follow-up, tumor biology, improving stage IV regimens, or the mere fact that anti-PD-1 therapy at the time of relapse may salvage the same patients that adjuvant therapy prevents is unclear. Notably, the trial design of KEYNOTE 054 sought to shed light on the benefit of adjuvant ICI versus treatment at time of relapse with patients assigned to the placebo arm eligible for cross-over to pembrolizumab at time of progression. In the updated 5-year analysis [78], the 5-year rate of RFS in the intention-to-treat population was 55.4% (versus 38.3% placebo arm), with a persistent RFS and DFMS benefit seen despite 72% of patients in the placebo arm receiving anti-PD-1-containing regimens for the treatment of locoregional recurrence and 64% for metastatic disease. Final OS data in this trial remain immature, but, to date, no clear OS benefit has been reported.

#### 2.11.2. Neoadjuvant Strategies

More recently, data have highlighted the benefit of neoadjuvant immunotherapy in the treatment of macroscopic stage III and resectable stage IV disease, with all studies highlighting the strong association of pathologic response rates, particularly the major pathologic response (MPR) and outcomes (RFS, DMFS) [79,80,81,82]. Compelling pre-clinical data suggest that neoadjuvant immunotherapy, compared with adjuvant therapy alone, stimulates a more effective and diverse immune response, leading to the improved eradication of micro-metastatic disease [83], with the SWOG1801 study establishing the benefit of peri-operative PD-1 inhibition versus adjuvant PD-1 inhibition alone [79]. In this randomized phase III study, one year of peri-operative pembrolizumab improved event-free survival (EFS), 72% versus 49%, compared to one year of adjuvant pembrolizumab, with no increase in trAEs.

Additional studies have demonstrated promising results with neoadjuvant ipilimumab (1 mg/kg) and nivolumab (3 mg/kg), fixed-dose nivo-rela [80,81,82], and the anti-TIGIT/PD-1 combination of vibostolimab plus pembrolizumab [70]. Based on the compelling results from the OpACIN-neo and PRADO trials [68,70], the phase III NADINA study is currently evaluating the efficacy of neoadjuvant ipi-nivo with a focus on personalizing the adjuvant therapy strategy based on the pathologic response (NCT04949113). The study plans to enroll 420 patients with recurrent or de novo macroscopic stage III melanoma. Patients will be randomized (1:1) to neoadjuvant or adjuvant treatment, with those in the neoadjuvant arm receiving two cycles of ipilimumab (80 mg) plus nivolumab (240 mg) followed by total lymph node dissection (TLND). In the case of a pathological partial response or non-response, surgery will be followed by adjuvant nivolumab or, in the case of *BRAF-V600*-mutant disease, adjuvant dabrafenib + trametinib. Patients in the adjuvant arm will undergo upfront TLND followed by nivolumab (480 mg) [84]. The primary endpoint is EFS. Results from the NADINA study, as well as others in the early-stage space [19,55], will continue to shape the ever-evolving peri-operative treatment paradigm.

## 3. Conclusions

In conclusion, the outcomes for patients with malignant melanoma have significantly improved over the last decade, with further practice-changing advancements seen across tumor stages over the last few years, including the approval of nivo-rela in front-line metastatic disease, encouraging data with TIL in anti-PD-1-resistant disease, and the benefit of peri-operative PD-1 inhibition in high-risk resectable disease. In the coming years, additional discoveries will ideally lead to expanded curative therapeutic options with an increased ability to personalize treatment for our patients.

## Figures and Tables

**Figure 1 cancers-15-04176-f001:**
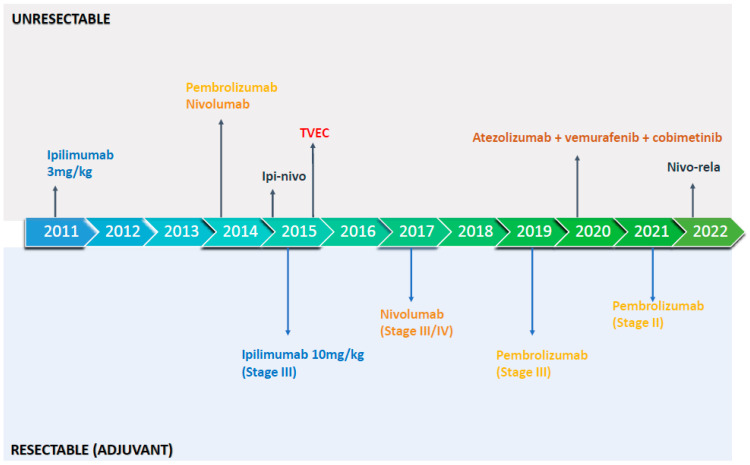
FDA-approved immunotherapy approaches in resectable and unresectable malignant melanoma. Ipilimumab: CTLA-4 inhibitor; Pembrolizumab: PD-1 inhibitor; Nivolumab: PD-1 inhibitor, Ipi-nivo (ipilimumab-nivolumab): CTLA-4 inhibitor plus PD-1 inhibitor; TVEC (Talimogene laherparepvec): injectable oncolytic virus; Atezolizumab-vemurafenib-cobimetinib: PD-L1 inhibitor plus BRAF inhibitor and MEK inhibitor, respectively; Nivo-rela (nivolumab-relatlimab): PD-1 inhibitor plus LAG-3 inhibitor.

**Table 1 cancers-15-04176-t001:** Summary of trials leading to therapeutic approvals in unresectable melanoma.

Metastatic/ Unresectable Trials	Active Treatment Arm (Randomization)	Control Arm	Primary Endpoint/s	OS Data	Treatment-Related Grade ≥ 3 Toxicity
Checkmate 066	Nivolumab 3 mg/kg Q2 weeks (1:1)	Dacarbazine 1000 mg/m^2^ Q3 weeks	OS	5-years OS: 39% vs. 17%	11.7 nivolumab vs. 17.6% dacarbazine
KEYNOTE 006	Pembrolizumab 10 mg/kg Q2 weeks or Q3 week (1:1:1)	Ipilimumab 3 mg/kg Q3 weeks × 4 doses	PFS and OS Median PFS: 8.4 months in combined pembro groups vs. 3.4 months in ipi group	5-years OS: 38.7% vs. 31%	17% pembro arms vs. 20% ipi
Checkmate 067	Ipilimumab (3 mg/kg) and nivolumab (1 mg/kg) Q3 weeks × 4 cycles followed by nivolumab maintenance or nivolumab 3 mg/kg (1:1:1)	Ipilimumab 3 mg/kg Q3 weeks × 4 doses	PFS and OS Median PFS 11.5 months ipi-nivo vs. 2.9 months for ipi alone	7.5-years OS: 48% ipi-nivo vs. 42% nivo vs. 22% ipi	59% ipi-nivo vs. 21% nivo vs. 28% ipi
RELATIVITY 047	Nivolumab-relatlimab (fixed dose, 480 mg nivo and 160 mg of rela) IV Q4 weeks (1:1)	Nivolumab	PFS: Median PFS 10.1 months vs. 4.6 months; 2-year PFS 39% vs. 29%	3-years OS: 56% vs. 48% **	21% nivo-rela vs. 11% nivo
IMspire150	Atezolizumab 840 mg IV Q2 weeks, vemurafenib 720 mg PO BID for 28 days, and cobimetinib 60 mg PO QD for 21 days (7 days off)	Vemurafenib 960 mg PO BID for 28 days and cobimetinib 60 mg QD for 21 days (7 days off)	PFS: Median PFS 15.1 months vs. 10.6 months	Median OS: 39 months vs. 25.8 months **	79% atezo-containing arm vs. 73% doublet
OPTiM	Talimogene laherparepvec intra-tumoral (2:1)	Subcutaneous recombinant GM-CSF	Durable response rate: 19.3 vs. 1.4%	Median OS: 23.3 months vs. 18.9 months	11.3% vs. 4.7%

PFS: progression-free survival; OS: overall survival; BID: twice a day; QD: once a day, GM-CSF: granulocyte-macrophage colony-stimulating factor. ** Non-statistically significant; numeric trend towards improved OS.

**Table 2 cancers-15-04176-t002:** Summary of trials leading to therapeutic approvals in resectable melanoma.

Adjuvant Trials	Active Treatment Arm (Randomization)	Control Arm	Melanoma Stage *	Updated RFS/EFS	Treatment-Related Grade ≥ 3 Toxicity
Checkmate 238	Nivolumab 3 mg/kg Q2 weeks for 1 year (1:1)	Ipilimumab 10 mg/kg Q3 week four doses and then every 12 weeks for 1 year	IIIB, IIIC, or IV	4 years RFS: 51.7% nivolumab vs. 41.2% ipilimumab	14.4 nivolumab vs. 45.9% ipilimumab
KEYNOTE 054	Pembrolizumab 200 mg Q3 week × 18 doses (1:1)	Placebo	IIIA (limited to lymph node metastasis of >1 mm) or stage IIIB or IIIC disease with no in-transit metastases	3-years RFS: 63.7% pembrolizumab vs. 44.1% placebo	14.7% pembrolizumab vs. 3.4% placebo
KEYNOTE 716	Pembrolizumab 200 mg Q3 week × 17 doses (1:1)	Placebo	IIB/C (TNM stage T3b or T4 with a negative sentinel lymph node biopsy)	2-years RFS: 81.2% pembrolizumab vs. 72.8% placebo	17% pembrolizumab vs. 5% placebo
Checkmate 76k	Nivolumab 480 mg Q4 weeks for 1 year (2:1)	Placebo	IIB/C (TNM stage T3b or T4 with a negative sentinel lymph node biopsy)	1-years RFS: 89% vs. 79% placebo	22% nivolumab vs. 12% placebo
SWOG1801 **	Peri-operative pembrolizumab: 3 cycles of neoadjuvant pembrolizumab 200 mg Q3 week followed by 15 cycles of adjuvant pembrolizumab (1:1)	Adjuvant pembrolizumab 200 mg Q3 week for 18 doses	IIIB/C/D and resectable IV	2-years EFS: 72% for peri-operative pembrolizumab vs. 49% adjuvant pembrolizumab	7% in neoadjuvant phase and 12% adjuvant phase for peri-operative pembrolizumab vs.: 14% adjuvant pembrolizumab only

* All studies with the exception of SWOG1801 were staged according to the criteria of the seventh edition of the American Joint Committee on Cancer cancer-staging manual (AJCC-7). ** Not yet approved. RFS: relapse-free survival, EFS: event-free survival; all trials had RFS/EFS as a primary endpoint.

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
