# Peer review of "Immunotherapy in Melanoma: Recent Advancements and Future Directions"

_cancers, 2023, doi:10.3390/cancers15164176_

Round 1

Reviewer 1 Report

This is a very comprehensive and up to date review of the clinical trial activity and outcome with ICI in melanoma.  The review offers not only the rationale for the trials but also explains clearly the implications of the outcomes for clinical practice.  The tables and figures are well presented and include the important toxicity data that is too often overlooked.  Efficacy of treatment is a driving criteria for the clinicians, but the treatment toxicity is also a major consideration for the patients. 

The only issue I had was with Figure 1.  The figure assumes a lot of knowledge on the readers' part. The names of the drugs and their targets are provided in the text, but if the authors add the names of the drugs with their targets to the figure legend it would greatly aid the readers understanding of the figure and how it relates to the rest of the review.   

Author Response

Thank you for your feedback. This information (Drug names and mechanism / target) was included in the figure legend. Please see attached in updated manuscript 

Reviewer 2 Report

This is a nice review of current treatment landscape for melanoma and key clinical trials that are ongoing. 

Informative for readers new to the field and nice summary of key trial data

I am wondering if TIGIT +PD1 can be another subcategory for review. 

Consider putting the section 2.5 as the last group as all other are treatment options, this is more on toxicity. 

Minor spelling errors to be reviewed. 

Overall easy to read, flows well. 

Author Response

Thank you for the comments. A section on TIGIT was included. 

Reviewer 3 Report

This review highlights recent advancements and future directions of immunotherapy in melanoma. There are only some minor issues, which should be addressed by the authors.

- The authors should be consisten by using metastatic/unresectable or unrescetable melanoma throughout the manuscript.

- Sometimes there is a space issue (eg. line 12) throughout the manuscript.

- IO should not be abbreviated in the heading of table 1. I also suggest to change the heading to "Summary of trials leading to the approval of treatment of melanoma in the metastatic/unresectable setting"

- Sometimes the name of the trials are mentioned including the NCT-registration number, sometimes it is not included. This should be consistent.

- There is a space missing after 11.4 (line 86).

-VEGF should not be abbreviated in the title of 2.3

- trAE should be explained in line 127, when first mentioning it.

-Tebentafusp should be abbreviated as "tebe" in line 130 and "tebe" should be consistently used throughout the following paragraph.

- In line 156 the authors should state the number of patients with percentage in brackets.

-TNF alpha-inhibition should be corrected to "TNF-alpha inhibition" in line 162.

- DLT should not be abbreviated in line 168.

-T-VEC is misspelled in line 175.

- In Chapter 2.8 some antigens are used abbreviated, some are not. This should be consitent.

- There is a space missing after 10.6 (line 296).

-Dab-tram should not be abbreviated, when first mentioning it (line 302).

-RFS should not be abbreviated (line 340).

-DFMS should not be  abbreviated  (line 357).

- I recommend to change the header of table 2 in analogy to table 1 to "Summary of trials leading to the approval of resectable melanoma". The SWOG1801 trial should be removed from this table and for the Checkmate 76k-trial a foot note should be added that it is awaiting approval.

- I suggest to devide the early stage-disease with subheadings into an adjuvant and neoadjuvant part.

- The KEYVIBE-010 trial should mentioned in the adjuvant trial setting as well as the results of the same combination in the neodadjuvant setting (presented at this years AACR).

Author Response

Thank you for the comments.

  1. The grammatical and formatting issues have been corrected.
  2. The figures / tables were edited to better reflect resectable / unresectable therapeutic approvals. 
  3. Information pertaining to anti-TIGIT in the adjuvant and neoadjuvant setting has been included

Notably re NCT trial number was included for all ongoing studies and deleted for those with published results.